# Determination of Mechanical Properties of Altered Dacite by Laboratory Methods

**Veljko Rupar [1],\*, Vladimir Čebašek [1] , Vladimir Milisavljević [1], Dejan Stevanović [1] and Nikola Živanović [2]**

1 Mining Department, Mining and Geology Faculty, University of Belgrade, 11000 Belgrade, Serbia; vladimir.cebasek@rgf.bg.ac.rs (V.Č.); vladimir.milisavljevic@rgf.bg.ac.rs (V.M.); dejan.stevanovic@rgf.bg.ac.rs (D.S.)
2 Department of Ecological Engineering for Soil and Water Resources Protection, Faculty of Forestry, University of Belgrade, 11000 Belgrade, Serbia; nikola.zivanovic@sfb.bg.ac.rs
\* Correspondence: veljko.rupar@rgf.bg.ac.rs; Tel.: +381-645494491

**Abstract:** This paper presents a methodology for determining the uniaxial and triaxial compressive strength of heterogeneous material composed of dacite (D) and altered dacite (AD). A zone of gradual transition from altered dacite to dacite was observed in the rock mass. The mechanical properties of the rock material in that zone were determined by laboratory tests of composite samples that consisted of rock material discs. However, the functional dependence on the strength parameter alteration of the rock material (UCS, intact UCS of the rock material, and $m_i$) with an increase in the participation of "weaker" rock material was determined based on the test results of uniaxial and triaxial compressive strength. The participation of altered dacite directly affects the mode and mechanism of failure during testing. Uniaxial compressive strength ($\sigma_{ci}^{UCS}$) and intact uniaxial compressive strength ($\sigma_{ci}^{TX}$) decrease exponentially with increased AD volumetric participation. The critical ratio at which the uniaxial compressive strength of the composite sample equals the strength of the uniform AD sample was at a percentage of 30% AD. Comparison of the obtained exponential equation with practical suggestions shows a good correspondence. The suggested methodology for determining heterogeneous rock mass strength parameters allows us to determine the influence of rock material heterogeneity on the values $\sigma_{ci}^{UCS}$, $\sigma_{ci}^{TX}$, and constant $m_i$. Obtained $\sigma_{ci}^{TX}$ and constant $m_i$ dependences define more reliable rock material strength parameter values, which can be used, along with rock mass classification systems, as a basis for assessing rock mass parameters. Therefore, it is possible to predict the strength parameters of the heterogeneous rock mass at the transition of hard (D) and weak rock (AD) based on all calculated strength parameters for different participation of AD.

**Keywords:** dacite; altered dacite; uniaxial compressive strength; triaxial compressive strength; composite samples

## 1. Introduction

Determining the mechanical properties of heterogeneous rock mass presents a major challenge in rock mechanics. Heterogeneous rock mass consists of two or more lithological members that have different properties. A limited number of studies have examined the mechanical properties of the heterogeneous rock mass that have dealt with examining uniaxial compressive strength (UCS) of composite samples. Tziallas et al. [1] were the only ones who conducted a triaxial test for only one volumetric participation of the weaker rock material.

Tziallas et al. [1] investigated flysch formations composed of sandstone and siltstone. Based on the proportion of certain lithological members in the rock mass, they formed composite samples of a certain ratio of sandstone and siltstone and simulated the actual condition in the rock mass in the laboratory. Duffault [2] published the study about modeling and simulation of heterogeneous rock mass by introducing the new term, "sandwich"

(individual or multiple) rock mass. Goodman [3] emphasized in his book that rock materials that consist of at least two lithological members and have different geomechanical properties are, in fact, the complex geotechnical problem. On the other hand, Z. Mohamed et al. [4] were the first ones who recently investigated UCS on composite samples. This study includes composites of sandstone and shale aimed at simulating tropical conditions that rule the area. Composite samples with the 10%, 20%, and 30% participation of weaker material of the overall sample height were prepared for the study, while gypsum was used as a bonding material between rock material discs. Liu et al. [5] examined composite samples comprised of rocks with different strengths and coals to determine the strength of the pillars of underground facilities formed in the rock mass. In their study, Berisavljević et al. [6] investigated how the strength of composite samples composed of sandstone and siltstone for building slopes on the state highway is changing. Liang et al. [7] were doing laboratory analysis on natural samples in layers consisting of salt rock and anhydrite to estimate the strength of rock material for building the storage of liquids, gases, and solid waste material. Greco et al. [8] investigated composite samples that contained different combinations of granite, marble, and limestone from Vicenza, from which the pillars and walls of the Cathedral were built to define strength parameters and mechanism of failure. Intact rock properties $\sigma_{ci}$ and $m_i$ for homogeneous rock mass should be carefully considered. Using the hard rock properties to determine the overall strength of the rock mass is not appropriate. On the other hand, using the intact properties of the weak rock is too conservative as the hard rock skeleton certainly contributes to the overall rock mass strength. Marinos and Hoek [9] suggested proportions of intact strength parameters $\sigma_{ci}$ and $m_i$ for estimating heterogeneous rock mass properties. Marinos [10] suggested modified proportions of values for heterogeneous rock types to be considered for the "intact rock" properties ($\sigma_{ci}$ and $m_i$) determination based on Marinos and Hoek [9].

This study aims to investigate the influence of variations in the volumetric participation of weaker rock material on the strength parameters of a heterogeneous rock mass, based on the results of laboratory tests of UCS and triaxial test of composite samples. This paper presents the methodology for determining the mechanical properties, UCS, and triaxial compressive strength or triaxial test on many composite samples with different participation of weaker rock material—altered dacite.

## 2. Geological Settings and Materials

Dacite deposit, "Ćeramide," belongs to the Rudnik–Ljig volcanic area, within which three larger masses of effusive rocks have been isolated. However, Sarmatian quartzite-dacite effusive with accompanying pyroclastics predominates within the volcanic area [11]. Dacite deposit is of simple geological structure and essentially contains three categories of Dacite rocks and Cretaceous flysch sediments. Nevertheless, dacite is grusified in the near-surface part, followed by a zone of altered dacite. The zone of altered dacite (AD) is yellowish due to the pronounced limonitization of biotite. After the altered dacite, there is compact dacite (D) of gray color, porphyry structure with pronounced large phenocrysts of feldspar.

Precise determination of the boundary between altered dacite and dacite was impossible through analyzing the available geological documentation, core mapping from the exploration boreholes, and the open pit slopes. However, the importance of the genesis and properties of the rock material in this zone (extrusive igneous rocks) concluded that the alterations in the rock mass occurred gradually (Figure 1). Subsequent interpretation of geological structure (Figure 1) singled out three zones: altered dacite, transitional zone, and compact dacite. The transition zone represented the interval in which there was a gradual alteration in the rock mass (dacite–altered dacite) and was defined from the mapped lower part of the altered dacite to the mapped upper part of the dacite.

| Elevation (m) | Depth (m) | Thickness (m) | Lithology | Lithological descrtiption | | |
|---|---|---|---|---|---|---|
| | | 4.0 | | Clay, sand, grus of dacit | | |
| 582.1 | 4.0 | 5.3 | | Altered dacit - cracked - tectonized | | AD |
| 576.8 | 9.3 | 6.0 | | Altered dacit - cracked - medium tectonized | | |
| 570.8 | 15.3 | 2.7 | | Weakly altered dacit - medium tectonized | | |
| 568.1 | 18.0 | 14.5 | | Dacit grey - weakly tectonized | | Transition zone |
| 553.6 552.9 | 32.5 33.2 | 0.7 | | Crashed dacit, mechanically crushed dacit | | |
| | | 5.0 | | Dacit grey - weakly tectonized | | |
| 547.9 | 38.2 | 5.3 | | Crashed dacit, mechanically crushed dacit | | |
| 542.6 | 43.5 | 9.2 | | Crashed dacit, mechanically crushed dacit | | |
| 533.4 | 52.7 | | | Compact dacit, grey | | D |

**Figure 1.** Lithological column of the exploration borehole.

　　Test samples D and AD undoubtedly show the same structural-textural characteristics, as well as the mineral composition. Only the intensities of secondary alterations, calcification, limonitization, and chlorination are different (Figure 2). The analyzed rock material samples have a holocrystalline porphyry structure with microcrystalline base mass and phenocrysts of quartz, plagioclase, sanidine, biotite, and hornblende. The accessory ingredients are apatite, zircon, and metallic minerals, while the secondary minerals are calcite, sericite, chlorite, and iron oxides/hydroxides. Mineralogical-petrographic analysis of D and AD samples showed that sanidine and quartz predominate in D. The strength parameters of the rock material are affected mainly by the content and size of quartz crystals. So, as the content of smaller quartz particles increases, the compressive strength increases. In the AD sample, the process of quartz resorption is pronounced, and biotite occurs in idiomorphic to hypidiomorphic laminated particles. In addition, the separation of metallic minerals is observed along cleavage planes. The presence of some "washed-up" laminated particles of biotite indicates that the rock was exposed to hydrothermal solutions. It was determined that AD has a yellow-pale brown to greyish color due to noticeable limonitization, which weakened the bond between individual mineral grains and significantly affected the strength of the rock material.

　　Knowledge of mineralogical-petrographic characteristics of rock material is essential when testing mechanical properties. The results of previous research show that the values compressive strength of rock material is significantly influenced by microstructural characteristics, the most important of which are: the participation of the main mineral elements [12], mineral grain size [13], arrangement of mineral grains and their shape [14], mutual contact between individual mineral grains [15], and level of alteration [16]. In the analyzed samples D and AD, it was observed that the mechanical properties of the rock

material are affected mainly by the mineralogical and petrographic characteristics of quartz and biotite.

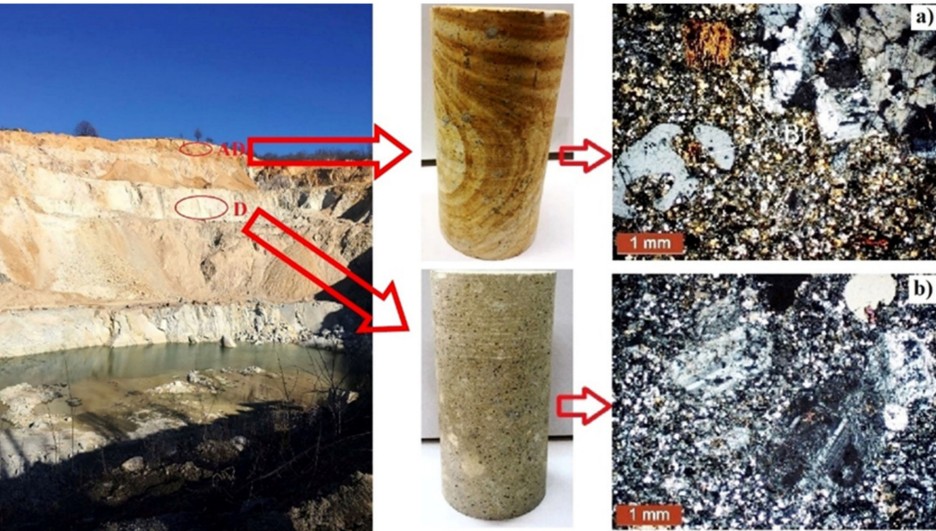

**Figure 2.** Analyzed open pit slope consisting of dacite with microscopic images of the material: (**a**) altered dacite and (**b**) dacite.

Exploration works were based primarily on exploratory drilling, geological mapping, and laboratory tests. During these studies, four boreholes were drilled, with a total drilling length of 535 m, and the average depth per borehole was 133 m [11].

### 3. Methodology of Laboratory Tests of Strength Parameters of Heterogeneous Rock Mass

Laboratory tests of UCS and triaxial test were performed on samples of homogeneous material and composite samples, for which formation blocks of undisturbed rock material (maximum dimensions of 30 cm × 30 cm × 30 cm) were used. Rock block samples of altered dacite were taken from the open pit bench at an elevation of 575 m a.s.l. and dacite at an elevation of 505 m a.s.l. (Figure 2), hermetically wrapped in polyethylene bags and transported to the laboratory. The composite specimens were prepared according to the procedure proposed by Tziallas et al. [1]. Cylindrical samples, 54 mm in diameter, of rock material were extracted by a high-quality laboratory coring machine with a diamond bit. The ends of the specimens and discs were then cut, polished, and shaped to the required height and diameter ratio with tolerances according to the American Society for Testing and Materials—ASTM [17] (Figure 3). Due to the previously shown procedure, monolithic rock material (100% D and 100% AD) specimens and discs of rock material were prepared. Discs of rock material were used to form groups of composite specimens that contained 10%, 30%, 50%, and 70% volumetric participation of weaker material (altered dacite) (Figure 4). It is essential to point out that no bonding material was used between the discs of rock material during the formation of composite samples. Using plaster to connect separate discs [4] was shown as inadequate due to plaster failure at lower loads, causing the discs to displace and introducing non-uniform stress distribution throughout the disc interfaces. Tziallas et al. [1] found that the use of plaster to connect separate discs is unnecessary, considering the pattern of the fracture surfaces and the fact that no displacements of the discs were observed, even at higher compressive loads. These findings are used in the present study.

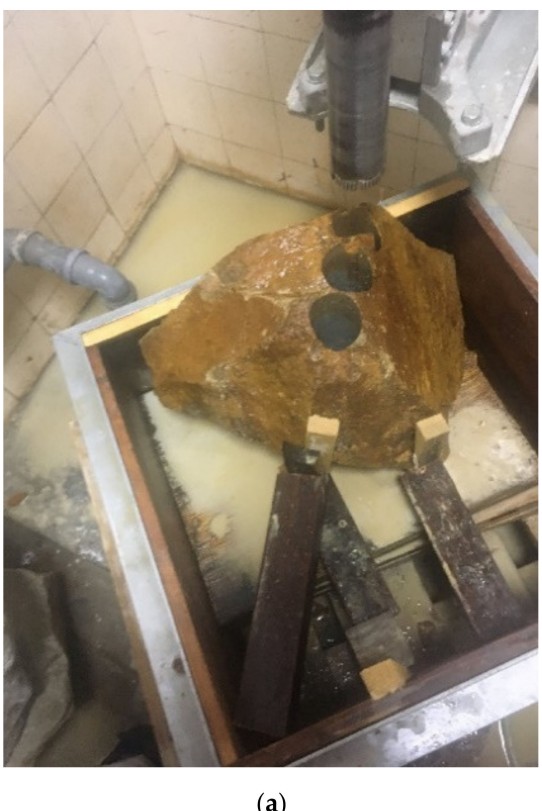

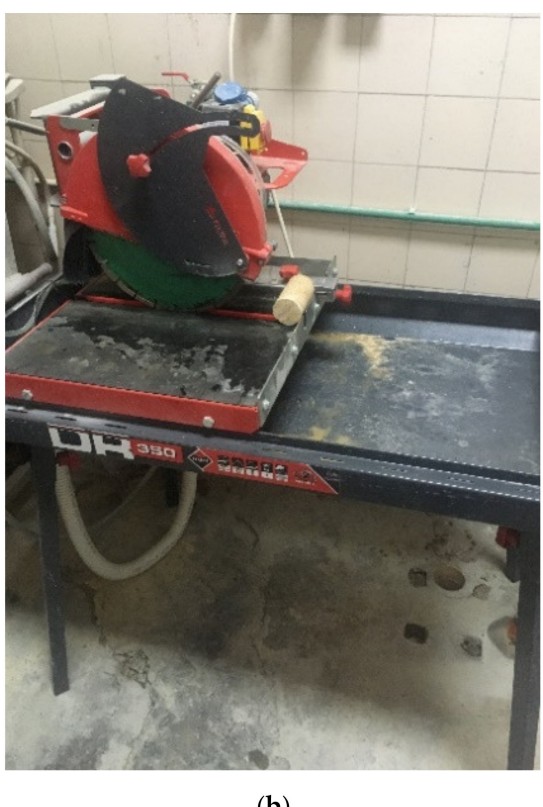

**(a)**                  **(b)**

**Figure 3.** Procedure for preparation of specimen for laboratory tests: (**a**) drilling process, (**b**) cutting process.

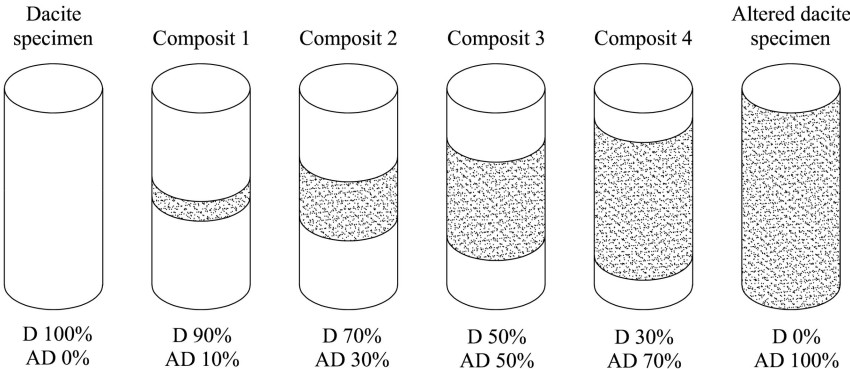

**Figure 4.** Scheme of tested samples preparation.

The dimensions of the specimens used in the UCS test differ depending on the recommendations and standards (ASTM and International Society for Rock Mechanics and Rock Engineering—ISRM). According to ASTM [17], the diameter of the specimen must not be less than 47 mm (ASTM) with a height to diameter ratio of 2–2.5:1, and the duration of the experiment should be 2–15 min. ISRM [18] suggested that the diameter of the specimen should not be less than 54 mm (ISRM), while the ratio of height and diameter should be 2.5–3:1, with the duration of the experiment from 5–10 min. According to the recommendations from previous studies [1,6], the diameter of the specimen for UCS testing was 54 mm, whereas the height and diameter ratio of the specimen ranged from 2–2.5:1. This research, in addition to the UCS tests, also conducted triaxial tests. Therefore, specimens with a diameter of 54 mm with a ratio of height and diameter of 2:1 were formed. In total, 60 specimens were prepared for laboratory tests, 30 for UCS tests, and 30 specimens for triaxial tests (Figure 5).

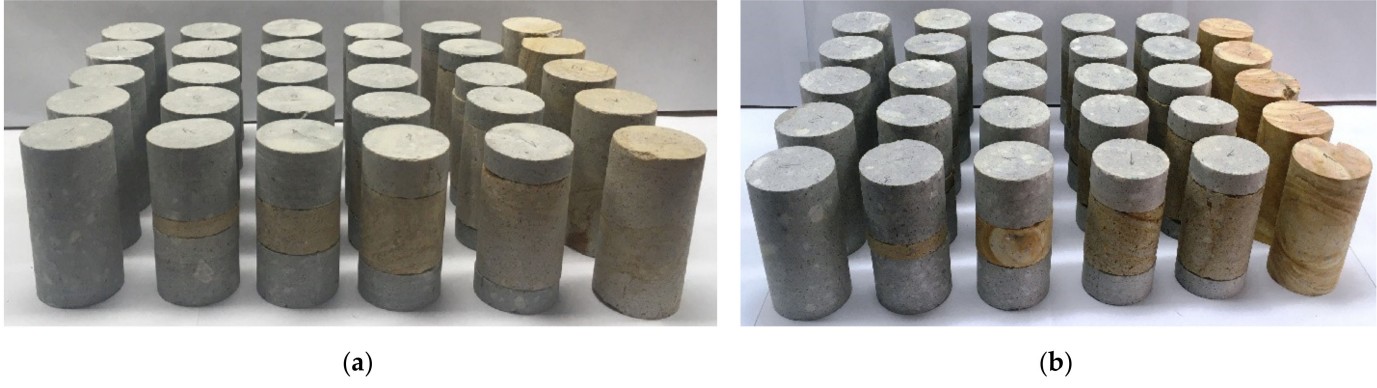

(**a**)                       (**b**)

**Figure 5.** Composite specimens prepared for testing: (**a**) UCS, and (**b**) triaxial test.

Nevertheless, UCS tests were performed on a hydraulic compression testing frame with sufficient pressure capacity and capable of applying a force perpendicular to the specimen base. UCS is calculated according to the following equation:

$$\sigma_{ci}{}^{UCS} = P/A \tag{1}$$

where: $P$—maximum force at the moment of failure, $A$—specimen's cross-section area.

Triaxial tests were carried out by the current ISRM recommendations for this type of test [19]. A triaxial test was performed for five different values of confining pressures ($\sigma_3$), namely, 5, 10, 15, 20, and 30 MPa. Considering that information on "in-situ" stress conditions was not available for this research, the values of confining pressures ($\sigma_3$) were defined according to the recommendations given by Hoek and Brown [20], where the values for the range of confining pressure are $0 < \sigma_3 < 0.5\sigma_{ci}$. During the test, the specimen first led to the hydrostatic stress state, i.e., with the increase in the confining pressure, the axial load of the sample increased in parallel until the hydrostatic state was reached for the set value of the sample of the confining pressure. During the test, the axial loads of the sample to achieve the hydrostatic state were: 11.5; 22.9; 34.4; 45.8; 68.7 kN. After reaching the hydrostatic state, a further increase in the axial load was outstretched until the failure while maintaining a constant confining pressure. The equipment for testing the triaxial test consisted of a standard Hoek triaxial cell diameter of 54 mm. The integral part consisted of pistons with spherical seats for load transfer, leading caps, and a rubber sealing sleeve (Figure 6a). Consequently, after placing the sample in the cell and connecting the manual confining pressure system (Figure 6b), the testing of rock material in triaxial conditions was performed.

The UCS and triaxial testing duration ranged from 5–10 min, with a stress rate of 0.1–0.3 kN/s. The specimens were tested at room temperature of $20 \pm 2\,°C$, and the time between field extraction of samples and the laboratory testing was no longer than 15 days.

The values of horizontal $\sigma_3$ and vertical $\sigma_1$ stress were obtained based on the registered values of the confining and axial load that led the sample to failure along the known surface of the specimen. Analysis and calculation of rock material strength parameters were performed by the generally accepted Hoek–Brown, and Mohr–Coulomb failure criteria.

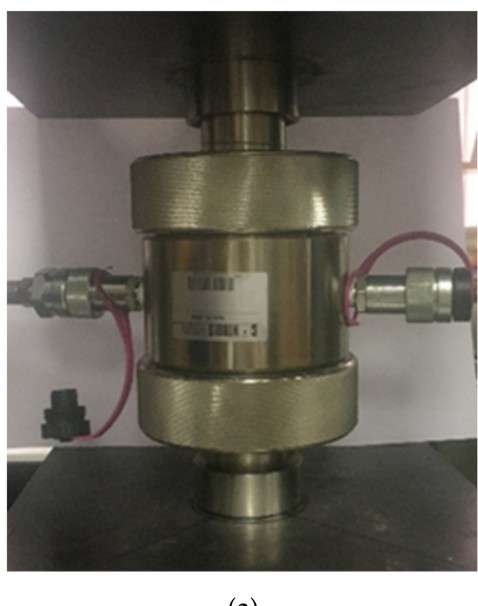

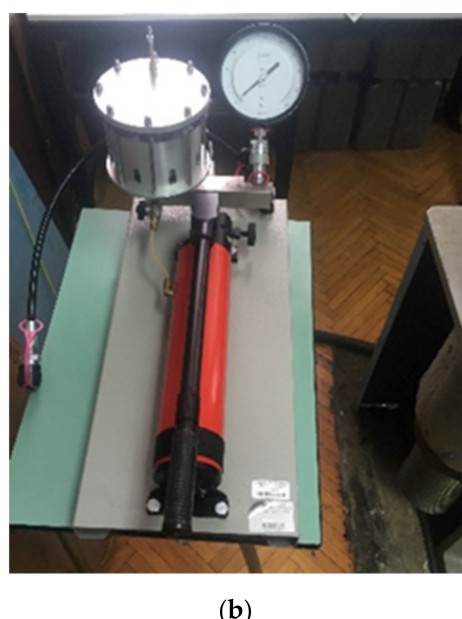

(**a**)
(**b**)

**Figure 6.** Triaxial test equipment: (**a**) Hoek cell and (**b**) manual confining pressure system for testing.

## 4. Results of Laboratory Strength Tests

### 4.1. Uniaxial Compressive Strength—UCS ($\sigma_{ci}{}^{UCS}$)

The unit weight (determined with paraffin) and the water content of the rock material were determined on samples of dacite and altered dacite. Unit weight tests were performed on 100 samples: 50 samples of dacite and 50 samples of altered dacite. The unit weight values in the altered dacite ranged from 24.00–24.82 kN/m$^3$ (mean value 24.37 kN/m$^3$), and in dacite 23.83–24.97 kN/m$^3$ (mean value 24.47 kN/m$^3$). The values of water content in altered dacite ranged from 7.52–8.87% (mean value 8.21%), and in dacite 6.82–7.94% (mean value 7.36%).

Test results of the UCS are shown in Table 1. The UCS values $\sigma_{ci}{}^{UCS}$ for dacite samples range from 100.74 to 106.77 MPa, with a mean value of 103.72 MPa. For the altered dacite samples, the values of UCS range from 32.70 to 34.85 MPa, with a mean value of 33.90 MPa. For composite samples, the minimum participation of altered dacite is 9.8%, while the maximum participation of altered dacite is 70.2%. Therefore, the UCS maximum value of $\sigma_{ci}{}^{UCS}$ = 57.87 MPa was for 9.8% AD, and the minimum value of $\sigma_{ci}{}^{UCS}$ = 33.31 MPa was for 70.1% AD.

Figure 7 shows specimens with characteristic failure patterns of monolithic and composite specimens for different participation of altered dacite. During the UCS testing of a dacite sample, brittle failure occurs, and the cracks are vertical, as shown in Figure 7a. In the prepared composites (Figure 7b–e), failure occurs in the weaker rock material, while on the examined specimens, it is clear that the failure spreads in dacite discs, which indicates that it is unnecessary to use any bonding material between the discs. After the fracture, the direction of the cracks spread through the entire composite sample [21], which is clearly shown in Figure 7b. The specimen of altered dacite (Figure 7f) has a laminar structure, and failure occurs in the direction of the lamination plane. The failure patterns and continuity of the failure surface indicate that bonding material at the interface zone between the discs of rock material is unnecessary.

**Table 1.** Test results of UCS.

| Material | Altered Dacite (%) | Uniaxial Compressive Strength $\sigma_{ci}^{UCS}$ (MPa) |
|---|---|---|
| Dacite | 0.0 | 100.74 |
| | 0.0 | 106.74 |
| | 0.0 | 103.21 |
| | 0.0 | 101.16 |
| | 0.0 | 106.77 |
| Composite sample | 10.1 | 59.16 |
| | 10.2 | 59.80 |
| | 9.8 | 57.87 |
| | 9.9 | 60.66 |
| | 10.0 | 58.08 |
| | 29.8 | 39.01 |
| | 30.1 | 38.58 |
| | 30.2 | 37.71 |
| | 29.9 | 35.58 |
| | 30.0 | 39.15 |
| | 50.3 | 34.37 |
| | 50.2 | 34.85 |
| | 50.4 | 36.01 |
| | 50.3 | 37.72 |
| | 50.1 | 36.87 |
| | 70.1 | 33.31 |
| | 70.0 | 33.99 |
| | 69.9 | 35.28 |
| | 70.1 | 35.28 |
| | 70.2 | 34.42 |
| Altered dacite | 100.0 | 34.24 |
| | 100.0 | 34.85 |
| | 100.0 | 34.29 |
| | 100.0 | 33.44 |
| | 100.0 | 32.70 |

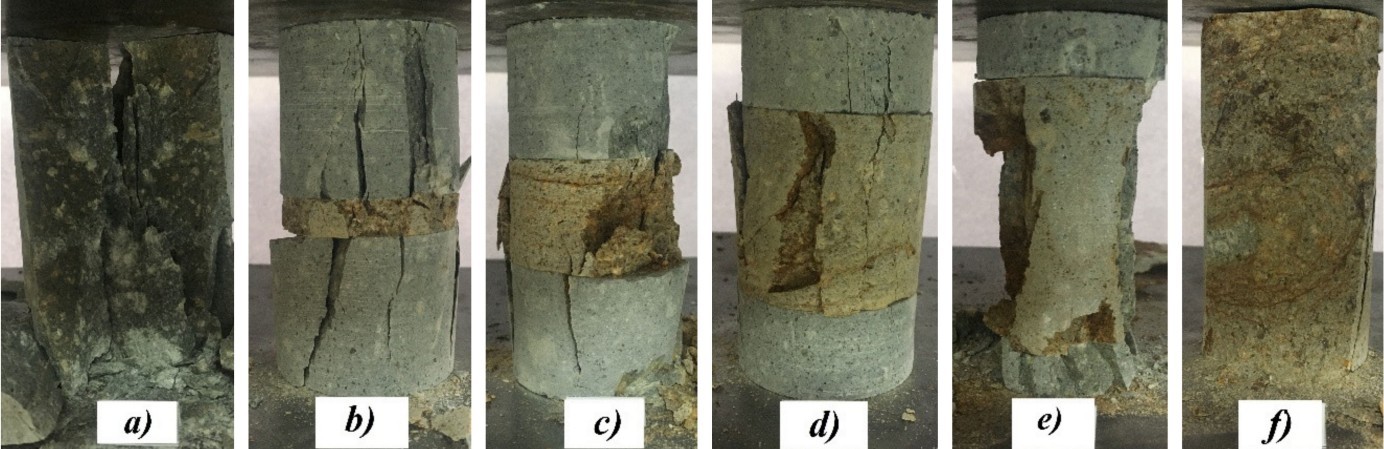

**Figure 7.** Specimens after testing—UCS: (**a**) 0% AD, (**b**) 10% AD, (**c**) 30% AD, (**d**) 50% AD, (**e**) 70% AD, (**f**) 100 % AD.

A diagram of the dependence of the UCS ($\sigma_{ci}^{UCS}$) on the volumetric participation of altered dacite (*AD*%) was drawn based on the test results (Figure 8). According to the previously performed research [1,6], the results were approximated by an exponential and linear function.

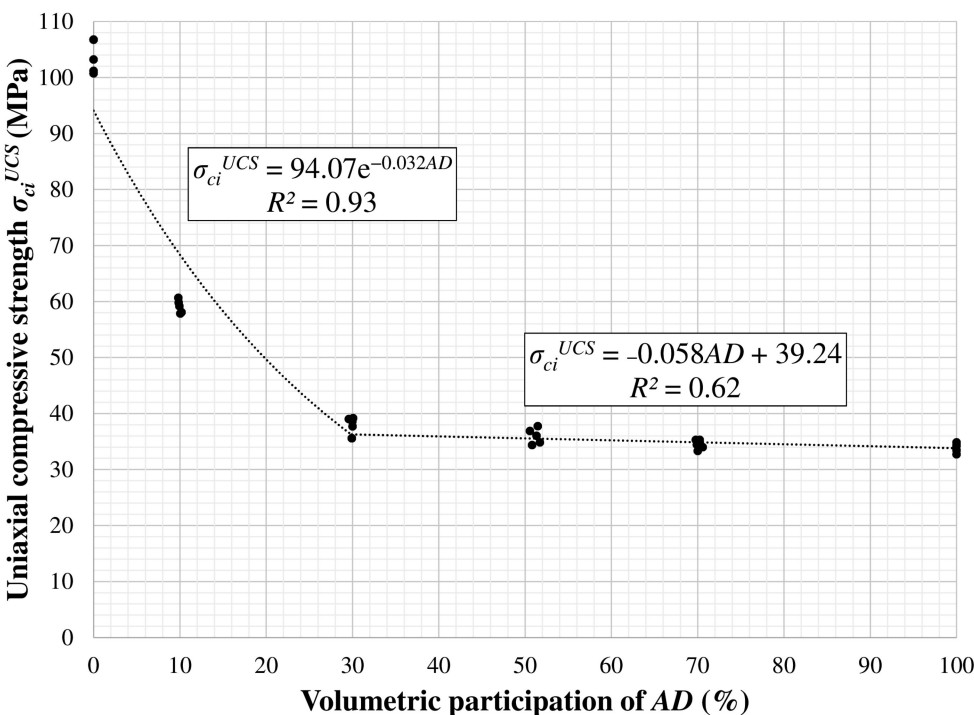

**Figure 8.** UCS results ($\sigma_{ci}{}^{UCS}$) for different participation of AD.

Dependence between $\sigma_{ci}{}^{UCS}$—AD% shown in the diagram is divided into two zones:

- the first zone for the values of altered dacite from 0% to 30% volumetric participation, where the value of $\sigma_{ci}$ decreases exponentially with increasing participation of altered dacite and is defined by the Equation (2) ($R^2 = 0.93$), and

$$\sigma_{ci}{}^{UCS} = 94.07 \times e^{-0.032 \times AD} \tag{2}$$

- the second zone for the values of altered dacite from 30% to 100% volumetric participation, where the value of $\sigma_{ci}$ decreases linearly with increasing participation of altered dacite and is defined by the Equation (3) ($R^2 = 0.62$)

$$\sigma_{ci}{}^{UCS} = -0.058 \times AD + 39.24 \tag{3}$$

The correlation coefficient in the first zone is $R^2 = 0.93$, which is classified as an excellent correlation [22]. However, the diagram clearly shows the line does not pass through a set of points for 0% and 10% of altered dacite. Therefore, there is a steep decline for values from 0–10%. On the other hand, the curve is milder for values from 10–30%. However, it is necessary to emphasize that further research to determine the values of UCS for 5%, 15%, and 20% volumetric participation of altered dacite (AD%) should be performed.

### 4.2. Triaxial Compressive Strength

The triaxial test was performed on groups of specimens ranging in the value of the volumetric participation of altered dacite by 0%, 10%, 30%, 50%, 70%, and 100%. During the test, the values of the confining load were $\sigma_3 = 5, 10, 15, 20,$ and 30 MPa. The compressive strength values were determined based on the previously mentioned conditions, representing the result of the triaxial test of the rock material and shown in Table 2.

**Table 2.** Test results of triaxial compressive strength.

| Material | Altered Dacite (%) | Confining Pressure $\sigma_3$ (MPa) | Compressive Strength $\sigma_1$ (MPa) |
|---|---|---|---|
| Dacite | 0.0 | 5 | 159.37 |
| | 0.0 | 10 | 212.29 |
| | 0.0 | 15 | 256.23 |
| | 0.0 | 20 | 294.51 |
| | 0.0 | 30 | 368.03 |
| Composite sample | 9.9 | 5 | 104.84 |
| | 9.9 | 10 | 150.08 |
| | 10.1 | 15 | 184.89 |
| | 9.8 | 20 | 214.90 |
| | 10.2 | 30 | 274.94 |
| | 29.6 | 5 | 74.82 |
| | 30.0 | 10 | 110.06 |
| | 30.0 | 15 | 140.08 |
| | 29.9 | 20 | 165.05 |
| | 30.1 | 30 | 214.90 |
| | 50.8 | 5 | 70.91 |
| | 51.7 | 10 | 103.05 |
| | 51.3 | 15 | 131.81 |
| | 51.5 | 20 | 157.04 |
| | 50.6 | 30 | 203.16 |
| | 70.0 | 5 | 68.05 |
| | 70.6 | 10 | 97.09 |
| | 70.2 | 15 | 126.13 |
| | 69.8 | 20 | 150.83 |
| | 69.9 | 30 | 195.91 |
| Altered dacite | 100.0 | 5 | 63.87 |
| | 100.0 | 10 | 92.07 |
| | 100.0 | 15 | 120.03 |
| | 100.0 | 20 | 144.89 |
| | 100.0 | 30 | 188.18 |

The appearance of specimens after testing with the characteristic failure patterns of composite specimens for different volumetric participation of altered dacite in the triaxial test is shown in Figure 9. Notably, Figure 9a–f shows that vertical and slightly angled cracks are formed during failure. In composite specimens, failure occurs in weaker rock material (Figure 9b–e). In specimens with 10% and 30% participation of altered dacite, the discs of dacite remain undisturbed, while in specimens with 50% and 70% participation of altered dacite, cracks are also observed on the discs of dacite.

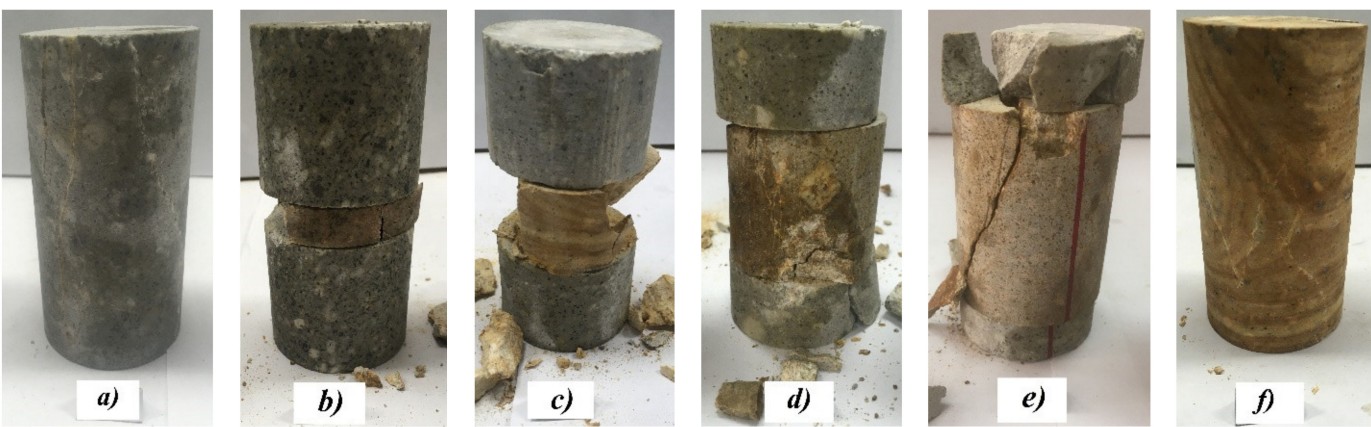

**Figure 9.** Specimens after testing—triaxial test: (**a**) 0% AD, (**b**) 10% AD, (**c**) 30% AD, (**d**) 50% AD, (**e**) 70% AD, (**f**) 100% AD.

The parameters of intact rock material for the Hoek–Brown failure criterion [23] were determined based on the test results of the triaxial test and mean values of UCS and for geological strength index values GSI = 100 and the disturbance factor D = 1 [24] using the program RocData, version: 5.013 (Rocscience Inc., Toronto, Ontario, Canada) (Table 3). The value of GSI = 100 was used to compare the uniaxial and triaxial compressive strength tests results performed in this study and compare the results of previous study of heterogeneous rock materials [1,4,6]. The actual values of GSI can be determined by detailed geological cores mapping from exploration boreholes and the open pit slopes, which makes it possible to determine the real values of the rock mass strength parameters—altered dacite, transition zone, and dacite. Furthermore, by analyzing the results of laboratory tests, the relationship between maximum and minimum principal stresses was determined. Figure 10 gives a relationship between maximum and minimum stress diagrams for different volumetric participation of altered dacite. It is clear that two envelopes stand out in the diagram: 0% and 10% of the volumetric value participation of altered dacite, while the envelopes of the value's volumetric participation of altered dacite for 30%, 50%, 70%, and 100% are very close.

**Table 3.** Strength parameters of Hoek–Brown failure criterion.

| Material | Intact UCS $\sigma_{ci}^{TX}$ (MPa) | Material Constant $m_i$ |
| --- | --- | --- |
| AD 0 | 101.350 | 31.581 |
| AD 10 | 58.000 | 29.014 |
| AD 30 | 37.329 | 25.267 |
| AD 50 | 34.397 | 24.166 |
| AD 70 | 33.408 | 22.467 |
| AD 100 | 32.306 | 20.871 |

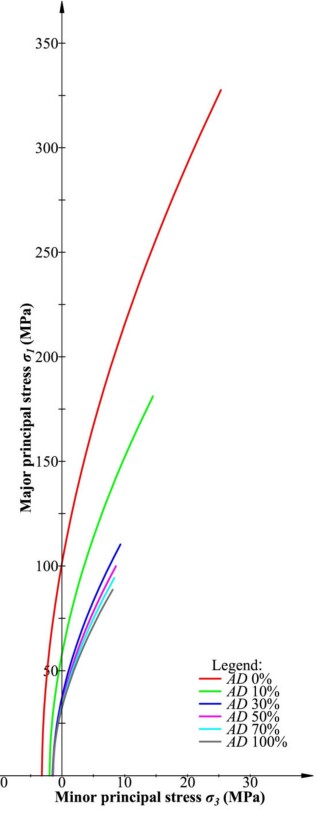

**Figure 10.** Relationship between maximum and minimum stress for different participation of AD.

Also, the analysis included relationships between shear and normal stresses that were constructed based on the Mohr circles of stress. This analysis determined the envelopes for all analyzed participation of altered dacite, which is shown in Figure 11.

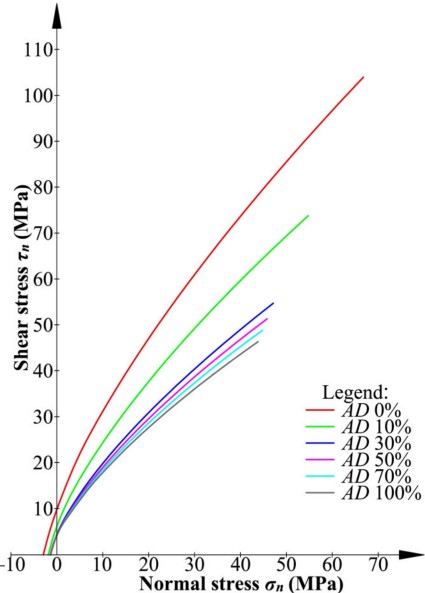

**Figure 11.** Relationship between shear and normal stresses for different participation of AD (Hoek–Brown failure criterion).

According to the relationship between shear and normal stresses for different participation of altered dacite (Figure 11) and the relationship between maximum and minimum principal stresses (Figure 10), it is clear that envelopes for 0% and 10% altered dacite volumetric participation stand out. In contrast, the envelopes for values volumetric participation of altered dacite from 30%, 50%, 70%, and 100% are very close.

The shear strength parameters of the Mohr–Coulomb failure criterion were calculated based on the measured laboratory data, which are shown in Table 4, while the comparative relationship is shown in the diagram (Figure 12).

**Table 4.** Shear strength parameters of the Mohr-Coulomb failure criterion.

| Material | Friction Angle $\varphi$ (°) | Cohesion $c$ (MPa) |
|---|---|---|
| AD 0 | 53.99 | 17.51 |
| AD 10 | 53.28 | 10.05 |
| AD 30 | 52.08 | 6.52 |
| AD 50 | 51.68 | 6.03 |
| AD 70 | 51.03 | 5.89 |
| AD 100 | 50.35 | 5.74 |

By inspecting the comparative presentation of shear strength parameters of the Mohr–Coulomb failure criterion (Figure 12), it is clear that, in this case, two lines stand out for values of altered dacite from 0% and 10% participation, while the lines for values of altered dacite 30%, 50%, 70% and 100% participation are very close.

The intact UCS values $\sigma_{ci}^{TX}$ and material constant $m_i$ were determined based on triaxial testing results for different values for different confining pressure values ($\sigma_3 = 5$, 10, 15, 20 and 30 MPa) and mean values of UCS ($\sigma_3 = 0$). The dependence of intact rock material compressive strength $\sigma_{ci}$ on the participation of altered dacite ($AD\%$) based on the analysis of laboratory test results is shown in Figure 13. An exponential and linear function

approximates the analysis results shown in the diagram (Figure 13), and the dependence $\sigma_{ci}^{TX}$—AD% is divided into two zones:

- the first zone for the value's volumetric participation of altered dacite from 0–30%, where the value of $\sigma_{ci}^{TX}$ decreases exponentially with increasing participation of altered dacite and is defined by the Equation (4), and

$$\sigma_{ci}^{TX} = 92.02 \times e^{-0.032 \times AD} \tag{4}$$

- the second zone for the value's volumetric participation of altered dacite from 30–100%, where the values for $\sigma_{ci}$ range from 32–37 MPa, and where the value of $\sigma_{ci}$ decreases linearly with increasing for the values participation of altered dacite and is defined by the Equation (5)

$$\sigma_{ci}^{UCS} = -0.067 \times AD + 38.59 \tag{5}$$

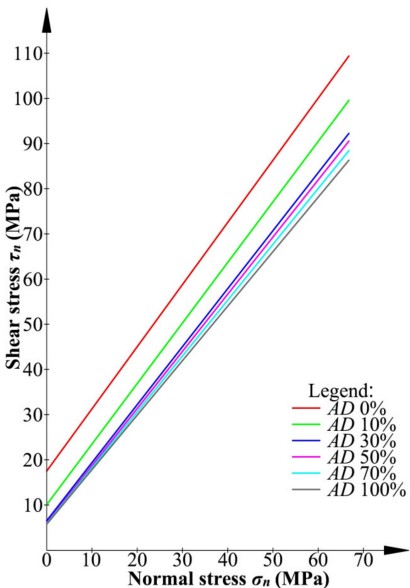

**Figure 12.** Relationship between shear and normal stresses for different participation of AD (Mohr-Coulomb failure criterion).

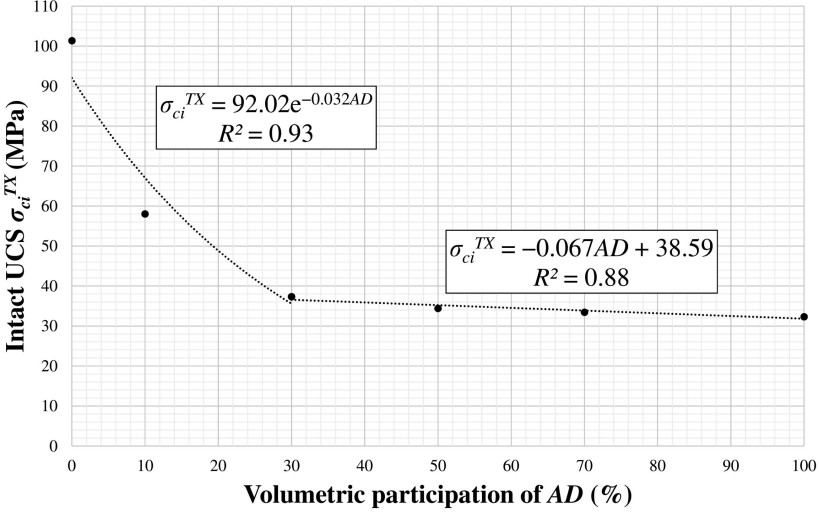

**Figure 13.** Values of intact UCS $\sigma_{ci}^{TX}$ for different participation of *AD*.

Figure 14 shows a diagram constructed by analyzing the value of the material constant $m_i$ depending on altered dacite ($AD\%$). It is clearly seen that two zones stand out in this diagram. In both zones, the value of the material constant decreases linearly with increasing participation of altered dacite, as follows:

- the first zone for the volumetric participation of altered dacite from 0% to 30%, where the value of $m_i$ decreases linearly with increasing participation of altered dacite according to Equation (6), and

$$m_i = -0.2072 \times AD + 31.383 \tag{6}$$

- the second zone for the volumetric participation of altered dacite from 30 to 100%, in which the values for $m_i$ decrease linearly with increasing participation of altered dacite according to the Equation (7)

$$m_i = -0.0643 \times AD + 27.213 \tag{7}$$

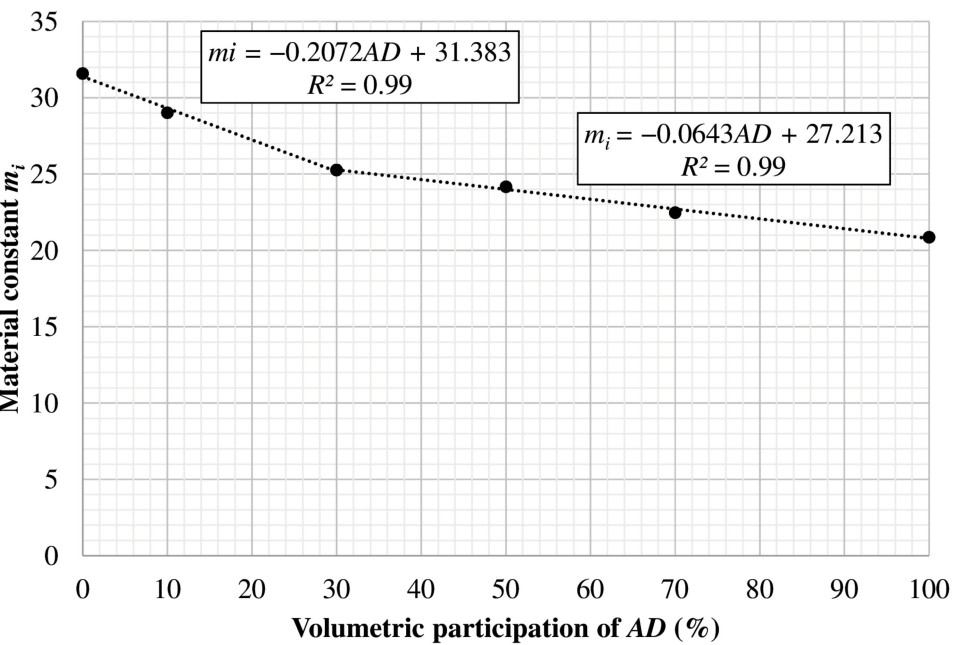

**Figure 14.** Values of the material constant $m_i$ for different participation of $AD$.

## 5. Discussion

This study includes laboratory UCS and triaxial tests of the composite specimens consisting of dacite and altered dacite, which have not been investigated so far. In previous studies, to make the composite specimens test results comparable, the intact UCS of each composite specimen ($\sigma_{ci}^{UCS}$) were normalized by the more robust material average intact UCS ($\sigma_{ci\,hard}^{UCS}$), giving the ratio $\sigma_{ci}^{UCS}/\sigma_{ci\,hard}^{UCS}$. This study determined the higher values of compressive strength of dacite and altered dacite rock material. The same approach was applied in this paper. Based on the previously presented analysis procedure, the comparison of the present results and the results of previous research and testing of composite samples [1,4,6] are shown in Figure 15.

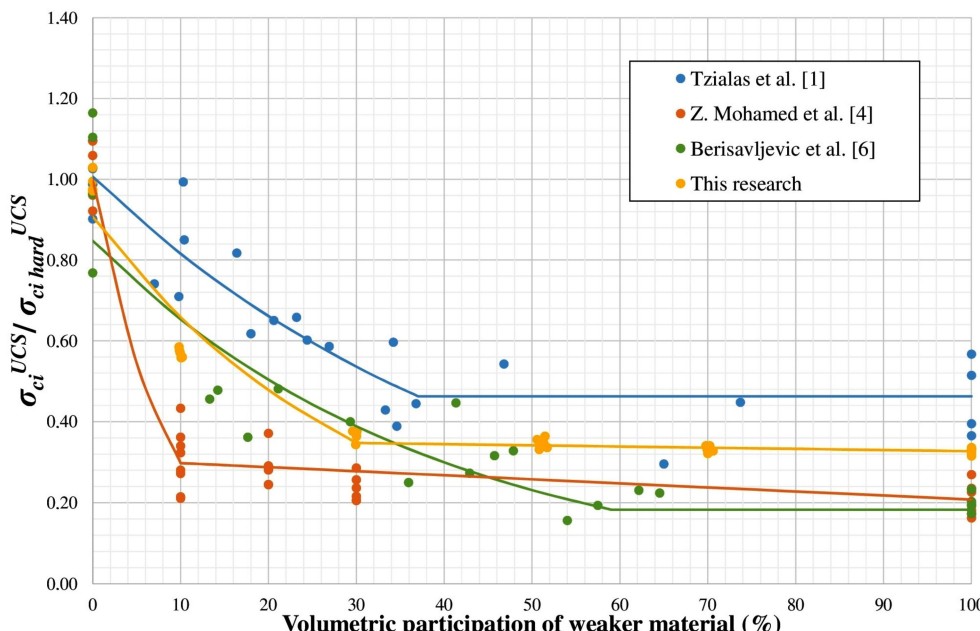

**Figure 15.** Comparison of the results $\sigma_{ci}{}^{UCS}/\sigma_{ci\ hard}{}^{UCS}$ with others studies.

Tziallas et al. [1] have emphasized that the strength decrease gradient directly depends on the hard and weak rock materials' intact UCS ratio. This study shows a ratio of $\sigma_{ci\ hard}{}^{UCS}/\sigma_{ci\ weak}{}^{UCS}$ = 2.2 and a moderate strength decrease gradient to about 38% of weaker rock material, leading the values to be approximately equal to intact UCS of weaker rock material. On the other hand, in Z. Mohamed et al. [4] study, the ratio was $\sigma_{ci\ hard}{}^{UCS}/\sigma_{ci\ weak}{}^{UCS}$ = 4.4 and strength decrease has a steeper gradient to about 10% of weaker rock material, where the strength decreases by about 70%. Afterward, it linearly decreases to 100% of the compressive strength of the weaker rock material. Finally, the Berisavljević et al. [6] study ratio was $\sigma_{ci\ hard}{}^{UCS}/\sigma_{ci\ weak}{}^{UCS}$ = 5. Although this ratio is higher than in two previous studies, the strength decrease has a lower gradient to about 59% of weaker rock material, where the strength decreases by about 70%, and later, the values were approximately equal to intact UCS of weaker rock material. In the present study, the ratio is $\sigma_{ci\ hard}{}^{UCS}/\sigma_{ci\ weak}{}^{UCS}$ = 3.1, and strength decrease has a steeper gradient up to 30% of weaker rock material, after which it almost linearly decreases to 100% of the value of weaker rock material.

The comparison between the results of this study with the previous one (Figure 16) shows that the strength decreases in Tziallas et al. [1] and Berisavljević et al. [6] have the same gradient in the range of 0% to 30%, and it is a slightly lower gradient compared to this study. However, Z. Mohamed et al. [4] study showed a much steeper strength decrease gradient, i.e., a large difference in strength was determined for specimens with 0 and 10% of the weaker rock material. Thus, the exponential trendline in Tziallas et al. [1] ends at 37% of the weaker rock material, and the ratio $\sigma_{ci}{}^{UCS}/\sigma_{ci\ hard}{}^{UCS}$ = 0.47. In Z. Mohamed et al. [4], it ends at 10% of the weaker rock material and the ratio $\sigma_{ci}{}^{UCS}/\sigma_{ci\ hard}{}^{UCS}$ = 0.30, while in Berisavljević et al. [6], the exponential trendline ends at 59% of the weaker rock material and the ratio $\sigma_{ci}{}^{UCS}/\sigma_{ci\ hard}{}^{UCS}$ = 0.18. In the present study, the exponential trendline ends at a value of 30% of the weaker rock material and the ratio $\sigma_{ci}{}^{UCS}/\sigma_{ci\ hard}{}^{UCS}$ = 0.34. After the stated values, in Tziallas et al. [1] and Berisavljević et al. [6], the ratio $\sigma_{ci}{}^{UCS}/\sigma_{ci\ hard}{}^{UCS}$ is constant up to 100% of the weaker rock material, while in Z. Mohamed et al. [4] study, the $\sigma_{ci}{}^{UCS}/\sigma_{ci\ hard}{}^{UCS}$ ratio decreases linearly. In the present study, the strength decrease is significantly lesser than the one determined by Z. Mohamed et al. [4], concluding the $\sigma_{ci}{}^{UCS}/\sigma_{ci\ hard}{}^{UCS}$ ratio is approximately constant.

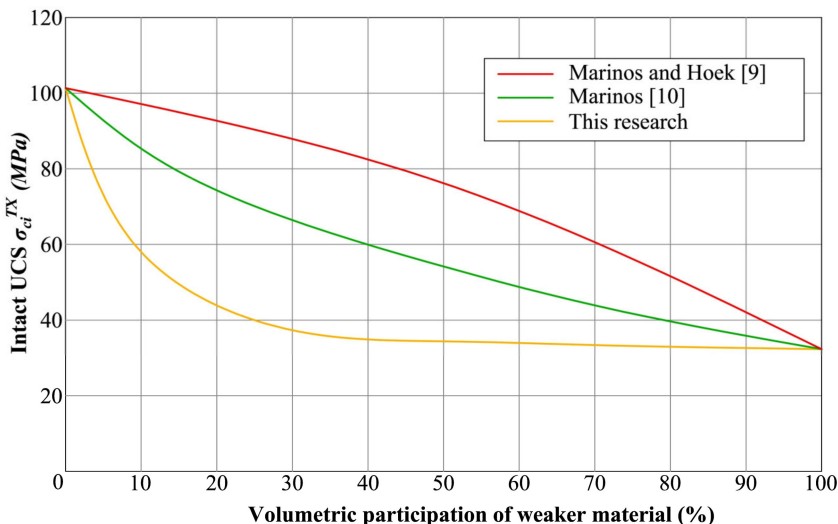

**Figure 16.** Comparison of the intact rock material UCS $\sigma_{ci}{}^{TX}$.

Marinos and Hoek [9] proposed a practical method for determining the intact strength properties of a heterogeneous rock mass. The authors suggested that it is not appropriate to use properties of stronger rock material to determine the overall strength of heterogenous rock mass. They also suggested that using the intact properties of the weaker rock material only is too conservative. Consequently, it is advised that the strength properties of the heterogeneous rock mass should be determined as the weighted average of $\sigma_{ci}$ and $m_i$ of intact strength properties of stronger and weaker rock material depending on their participation in the rock mass. In addition, when estimating the strength parameters of the heterogeneous rock mass, it is proposed to use the stronger rock material strength values reduced by 20% to 60%, depending on its participation and the structure of the heterogeneous rock mass. Based on these recommendations, the intact strength properties of a heterogeneous rock mass $\sigma_{ci}$ and $m_i$ were determined for 0%, 50%, and 100% of the weaker rock material (altered dacite) volumetric participation. The dependence of the $\sigma_{ci}$ and $m_i$ parameters and the altered dacite volumetric participation is presented in Figures 16 and 17.

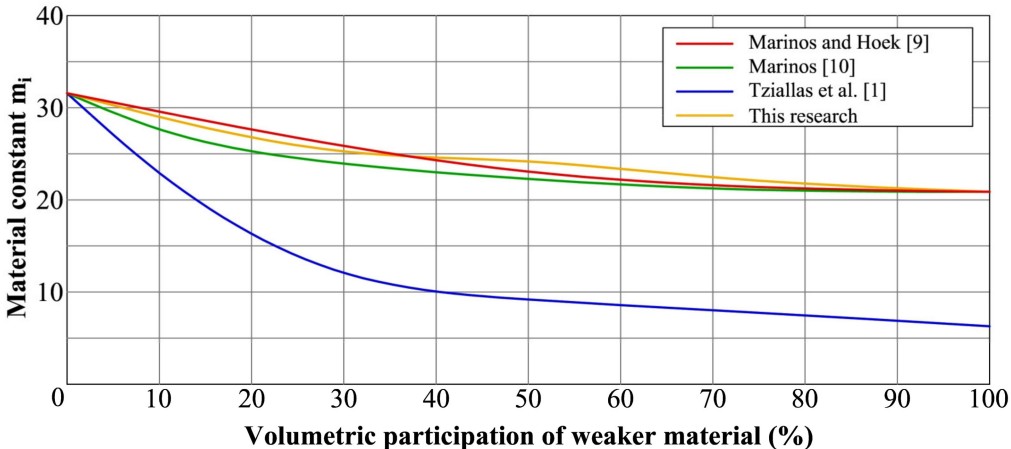

**Figure 17.** Comparison of the value material constant $m_i$.

Marinos [10] expanded and modified the original practical method for determining the intact strength properties of a heterogeneous rock mass. In this method, too, the strength properties of a heterogeneous rock mass $\sigma_{ci}$ and $m_i$ are determined as the weighted average of $\sigma_{ci}$ and $m_i$ of intact strength properties of stronger and weaker rock materials depending on their participation in the rock mass. Accordingly, the recommendation

was to use the stronger rock material strength values reduced by 10% to 40%, depending on its participation and the structure of the heterogeneous rock mass when estimating the strength parameters of the heterogeneous rock mass. The values of intact strength properties of a heterogeneous rock mass $\sigma_{ci}$ and $m_i$ for 0%, 10%, 30%, 50%, 70%, and 100% of the weaker rock material (altered dacite) participation determined based on these recommendations are shown in Figures 16 and 17.

Tziallas et al. [1] performed triaxial laboratory tests of a group of composite specimens with around 16% of the weaker rock material participation. Based on the performed test results, the intact UCS $\sigma_{ci}$ and constant $m_i$ were determined according to the Hoek–Brown failure criterion. In addition, the authors proposed an equation that can be used to estimate the values of the constant $m_i$ depending on weaker rock material volumetric participation expressed as Equation (8). According to the proposed dependence for $m_i$, for the present study, the dependence of the constant $m_i$ and volumetric participation of the weaker rock material is as Equation (9). The values of the constant $m_i$ for 0%, 10%, 30%, 50%, 70%, and 100% of the weaker rock material (altered dacite) volumetric participation determined based on the previously presented equation are shown in Figure 16.

$$m_i = 17 \times e^{0.022 \times SL} \tag{8}$$

$$m_i = 31.581 \times e^{0.032 \times AD} \tag{9}$$

where: *SL*—volumetric participation of siltstone [1], *AD*—volumetric participation of altered dacite.

By analyzing the comparison of the present study results and the recommendations given in the presented literature [9,10], it can be observed that the intact UCS $\sigma_{ci}$ determined in these studies are significantly lower than the values determined based on the Marinos and Hoek [9] and Marinos [10] practical methods (Figure 16). The previous conclusion indicates that the application of these practical methods for estimating the $\sigma_{ci}$ has certain limitations and that both methods overestimate the values of intact UCS $\sigma_{ci}$. On the other hand, the constant $m_i$ determined by the present study significantly correlates to the values determined according to practical methods given by Marinos and Hoek [9], and Marinos [10] (Figure 17). Therefore, applying both of these practical methods for constant $m_i$ estimation is justified, so in this way, one can obtain a realistic estimation of the heterogeneous intact rock mass constant $m_i$. The constant $m_i$ determined using the procedure proposed by Tziallas et al. [1] is significantly lower than the values determined by the present study.

Marinos and Hoek [9] proposed values of the constant $m_i$ for intact rock by the rock group, while for dacite, the recommended value for $m_i = 25 \pm 3$. In the present study, the values of constant $m_i$ varied from 20.871 to 31.581. However, the overall interval of constant $m_i$ values refers to dacite and altered dacite and the transition zone between these two rock materials. Based on the dependence of the constant $m_i$ and the altered dacite (*AD*%) volumetric participation (Figure 17), the overall interval of the $m_i$ value can be divided into two intervals. The first interval includes values for 0% to 30% of altered dacite (*AD*%) volumetric participation, where the constant is $m_i$ ranges from 20.871 to 25.267. However, the second interval includes values for 30% to 100% of altered dacite (*AD*%) volumetric participation, where the constant $m_i$ ranges from 25.267 to 31.581.

## 6. Conclusions

In order to define the influence of lithological heterogeneity on the Hoek–Brown failure criterion parameters, uniaxial compressive strength and triaxial tests on specially prepared composite specimens of rock material were performed. Composite specimens were meant to represent the transitional zone. They also comprised dacite and altered dacite discs with different volumetric (thickness) ratios. Consequently, monolith specimens consisting of dacite and altered dacite were made.

According to the test results, both UCS-$\sigma_{ci}^{UCS}$ and intact UCS-$\sigma_{ci}^{TX}$ dependence on altered dacite (*AD*%)-volumetric participation can be divided into two zones. The first zone includes test results of dacite specimens (0% AD) and composite specimens with 10% and 30% volumetric participation of altered dacite. In this zone, the UCS-$\sigma_{ci}^{UCS}$ and intact UCS-$\sigma_{ci}^{TX}$ decreases exponentially from the initial value (100%) to around 37%. In the second zone, the composite specimens with 30% and more volumetric participation of altered dacite show slightly reduction of UCS-$\sigma_{ci}^{UCS}$ and intact UCS-$\sigma_{ci}^{TX}$ and are approximately equal to the altered dacite UCS-$\sigma_{ci}^{UCS}$ and intact UCS-$\sigma_{ci}^{TX}$ (Figure 18).

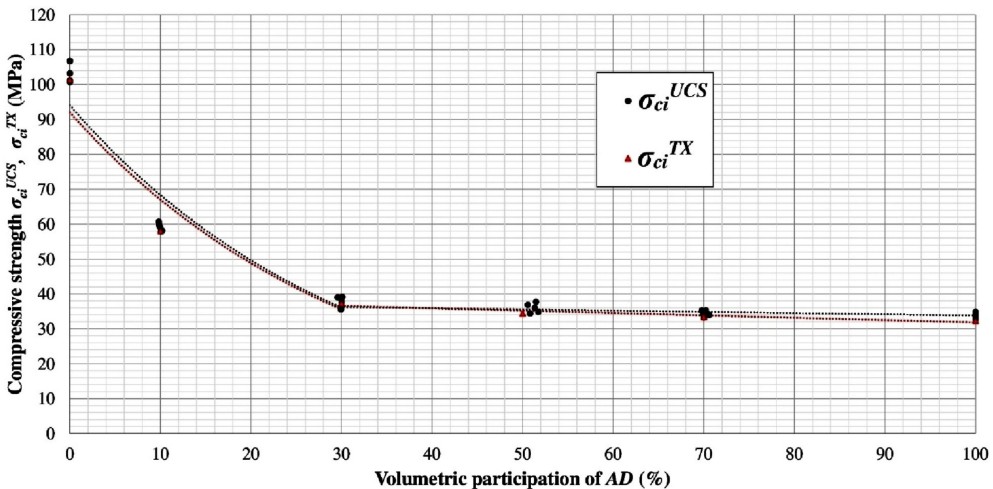

**Figure 18.** Comparison of the UCS-$\sigma_{ci}^{UCS}$ and intact UCS-$\sigma_{ci}^{TX}$ value.

The gradient of the strength decrease (as indicated by the $\sigma_{ci\,hard}^{UCS}/\sigma_{ci\,weak}^{UCS}$ ratio and $\sigma_{ci\,hard}^{TX}/\sigma_{ci\,weak}^{TX}$) did not match the findings of Tziallas et al. [1] and Berisavljević et al. [6]. The reasons for this should be sought in the effect of rock material mineralogical-petrographic composition and micro-heterogeneity (heterogeneity on the level of mineral grains of individual rock types).

The values of the constant $m_i$ are calculated based on triaxial tests results, and the overall interval of constant $m_i$ values could be generally expressed as $m_i = 26 \pm 6$. The overall interval of constant $m_i$ values refers to dacite and altered dacite and the transition zone between these two rock materials and can be divided into two intervals. Namely, for the first interval (0% to 30% of AD), it is possible to define the value of the constant $m_i = 28 \pm 3$, which refers to dacite. However, for the second interval (30% to 100% AD), the constant value is $m_i = 23 \pm 3$, which refers to altered dacite. The presented values of constant $m_i$ indicate that a detailed assessment of the rock material alteration degree is essential when estimating the constant $m_i$ value.

The suggested methodology for determining heterogeneous rock mass strength parameters and presented rock material strength parameters depends on determining the influence of rock material heterogeneity on the values UCS-$\sigma_{ci}^{UCS}$, intact UCS-$\sigma_{ci}^{TX}$, and constant $m_i$. However, intact UCS-$\sigma_{ci}^{TX}$ and constant $m_i$ dependences define more reliable rock material strength parameter values that can be used, along with rock mass classification systems, as a basis for assessing rock mass input parameters necessary for geotechnical stability analysis and detailed design solutions.

**Author Contributions:** Conceptualization, V.R.; methodology, V.R. and V.Č.; validation, V.M., D.S. and N.Ž.; formal analysis, V.R. and V.Č.; investigation, V.R., V.Č., V.M., D.S. and N.Ž.; resources, V.R. and V.Č.; data curation, V.M., D.S. and N.Ž.; writing—original draft preparation, V.R.; writing—review and editing, V.Č.; visualization, V.M., D.S. and N.Ž.; supervision, V.Č. All authors have read and agreed to the published version of the manuscript.

**Funding:** This research received no external funding.

**Data Availability Statement:** Not applicable.

**Conflicts of Interest:** The authors declare no conflict of interest.

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
