# Peer review of "Determination of Mechanical Properties of Altered Dacite by Laboratory Methods"

_minerals, doi:10.3390/min11080813_

Round 1
Reviewer 1 Report
I read the paper carefully and I find it interesting enough for publishing in Minerals Journal. The topic is not novel, but there are few complex research on one rock only in the literature, and none for diacite (frankly, quite a rare rock in mining and tunelling engineering). The most interesting is the quantitative determination of the reduction of mechanical parametrs of this rock. Everybody knows, that whethering reduces it, but nobody knows how much.
I have three general comments to the paper:
- Abstract should be more informative. Give some results here, please.
- The effect of the wheathering is the change of mineral contribution. You give the short description in lines 86-97. But it would be essential to compare the composition of diacite and altered diacite. That is the reason that your results don't match the results of other researchers commented at the end of the paper. It's just about the level of wheathering and its intensity.
- During the analysis in page 11 you give GSI value 100, so the highest possible at all for the analysis! The real value for diacite and much less for altered diacite is lower. Even for intact igneous rock you never get 100! This value influences on the results of rock parameters calculation, which are high and false. The comment here is needed or another values (to differentiate between these two rocks: altered and non-altered).
The minor remarks are as follows:
- line 50 -"siltit"??
- line 70/71 - "grusified" sounds weird
- line 93 - "other" - type error
- line 107 - why not to say "by drilling rig with a diamond crown"
- Fig 6 and 7 - I suggest to reduce one of them. They show the same cylinder.
- lines 243/244 - I suggest to put dot after the "(Fig. 13) and cross the rest of the sentence
- Fig. 13 - H-B "strength" envelopes?
- line 302 - I would cross "with this in mind"
- Fig. 18 - you write about sigma proportion of 2.2 (Tzialllas) or even 5 (Berisavljević) but there is max. 1.2 in the chart. It is unclear.
- Conclusion - shroten this chapter as you repeat in details some facts given in the previous paragraphs.
Give the mechanical parametrs (e.g. σi , mi ) in italic.
Author Response
Dear reviewer,
the answers to your remarks and suggestions by items are attached.
Best regards
Authors

Reviewer 2 Report
1) The purpose of the study should be given more clearly in the abstract.
2) The difference between the studies on decomposition in the introduction and the difference between this study and its target should be supported by more literature.
3) It would be better if the experimental methods and findings section could be shortened.
4) The results section should be summarized to include the most important results rather than all results of the study.
Author Response

(The authors gave the same response as above.)

Reviewer 3 Report
Please see my comments in the attached manuscript file.

Author Response

(The authors gave the same response as above.)

Reviewer 4 Report
The paper addresses the determination of the material strength in heterogeneous rock masses. It is based on a series on lab experiments with specimens composed of dacite and altered dacite considering various proportions of the 2 components. The results of uniaxial and triaxial compression tests are presented, allowing the determination of the Hoek-Brown and Mohr-Coulomb strength parameters as a function of the specimen composition. The experimental data presented in the paper provides useful information for the interpretation of characterization studies of heterogeneous rock masses. The presentation is clear and complete. The reviewer recommends the publication with some corrections.
1 – There is some redundancy in the presentation of results. If necessary, the paper could be shortened. For example, Fig. 13, with the Mohr circles, could be eliminated without loss of information. Also, some of the plots in Fig. 11 could be removed, since the match of the curve and points is of similar order in all cases. Fig. 12 gives a better overall behavior using the same scale for all cases.
2 – The text is generally well-written, but there are a few unclear passages that should be corrected. for example:
Line 298: “which has not been a part of the research so far”. Perhaps “which has not been investigated so far”.
Line 302: ”in the following paper” is unclear.
Line 446: “object” ?
Author Response

(The authors gave the same response as above.)

Round 2
Reviewer 3 Report
Dear Authors,
Congratulations on your excellent work. Thank you for the revisions to the manuscript. There are some minor editorial issues which I trust will be resolved during the publication process (e.g. citation style such as "Z. Mohamed et al." in the text is not acceptable).